# Are There Differences in Video Gaming and Use of Social Media among Boys and Girls?—A Mixed Methods Approach

**DOI:** 10.3390/ijerph18116085

**Published:** 2021-06-04

**Authors:** Marja Leonhardt, Stian Overå

**Affiliations:** Norwegian National Advisory Unit on Concurrent Substance Abuse and Mental Health Disorders, Innlandet Hospital Trust, P.O. Box 104, 2381 Brumunddal, Norway; Stian.Overa@sykehuset-innlandet.no

**Keywords:** adolescence, gender, gaming, survey, thematic analysis, group interviews, mixed methods

## Abstract

Gaming is widespread among adolescents and has typically been viewed as an activity for boys. There are however a growing number of female gamers and we need to learn more about how gender affects gaming. The aim of this study is to both quantify gaming among Norwegian adolescents and explore how gender differences are perceived. A mixed method approach was used to capture gaming experiences among boys and girls. Survey data (*N* = 5607) was analyzed descriptively, and five focus groups were conducted, applying thematic analysis. Statistics showed that boys from the age of 14 use video games up to 5 times more than girls, while girls are much more on social media. From the focus groups, we found that boys did not view social media as socially significant as gaming and that there is a greater social acceptance of gaming among boys than among girls. Gender differences in video gaming are not necessarily a problem per se, as they may reflect gender-specific motivations and interests. However, the study also finds that girls feel less encouraged than boys to play video games due to different gender-related experiences of video gaming. Therefore, gendered barriers in video gaming must be explored in future research.

## 1. Introduction

Video games have been played for decades, influencing the daily lives of people and relationship to technology. In the last 40 years, video games have moved from amusement arcades (1970s) to home consoles and computers (1980s–1990s), and more recently to mobile devices (2000s) and lastly to sports stadiums due to the popularity of electronic sport, eSports (2010s) [1]. Brilliant et al. describe video gaming as “the experience of playing electronic games, which vary from action to passive games, presenting a player with physical and mental challenges” [2]. According to the analysis company Newzoo, 2.8 billion people will play video games during 2021, and by the end of the year, the global gaming market will be worth USD 189.3 billion [3]. The popularity of video games has sparked considerable debate in Norway and abroad about how gaming may affect children and adolescents [4].

As pointed out by Lobel et al. most research on video gaming has focused on its potential negative impacts, such as addiction, violence, depression, and other mental health issues [5]. Similarly, the media have often portrayed gaming as a harmful activity. For example, the Norwegian media linked the terrorist attacks by Breivik in 2011 to the fascination with video games, such as Call of Duty and World of Warcraft [6]. Harmful gaming, described as gaming disorder, is listed in the 11th version of the International Classification of Diseases (ICD-11) [7]. Much valuable research on the possible negative aspects of gaming exists, which is indeed essential knowledge for the prevention and treatment of problematic gaming. However, as suggested by Billieux et al., research has often focused on the small minority of gamers that experience severe problems [8].

In recent years, a significant body of studies has begun to investigate gaming’s potential positive side. Research has shown that gamers can experience considerable benefit, such as pleasure and enjoyment [9], meaningful and lasting relationships [10], opportunities for personal growth and improved relational skills [11], and a sense of belonging [12]. In line with this new body of research, we argue for an open and exploratory approach to the everyday experiences of video gamers, focusing on both the possible positive and negative aspects of video games.

Video gaming has long been perceived as mainly a territory for adolescents and males. A systematic review on gender differences in online gaming by Veltri et al. [13] reports both gender similarities and differences in gaming behavior. Overall, the literature shows that male gamers start playing video games earlier in life, play more frequently, and spend more time playing video games than female gamers [13]. However, the gaming landscape is changing, and several studies show a recent increase in the numbers of female gamers. A survey by the Norwegian Media Authority reveals that the proportion of girls aged 9–18 who play video games regularly increased from 2018 to 2020. In 2020, 96% of boys and 76% of girls reported playing video games regularly, while the corresponding figures were 96% and 63% in 2018 [14].

Since video gaming is more prevalent among boys than girls [15,16], the literature has particularly focused on male adolescence. As McLean and Griffiths [17] argue, female gaming is a relatively under-researched area. However, several studies have indicated that gender is essential for gaming experiences [18,19,20]. Condis argues that, despite the supposedly disembodied nature of online games, video game players experience gender, sexuality, and race concurrently [21]. Since most studies have used male participants, more research on how gender is related to gaming practices is needed. As the number of female video gamers is growing, it is essential to understand their motives and behavior. The Norwegian action plan for preventing problematic gaming and gambling specifically requests more research on gender differences in gaming behavior [22].

In this light, the purpose of this study is to explore gender differences in various aspects of video gaming. We believe that exploring gender disparities is vital because gender is a fundamental characteristic that inspires the behavior and societal roles of men and women. The present study aims to investigate [1] how gender effects video gaming behavior in Norwegian adolescents and [2] how secondary school students in Norway perceive and understand gender differences in video gaming.

## 2. Materials and Methods

### 2.1. Mixed Methods

Our study population was children and adolescents in Norway aged 11–19 years. We used a mixed-method approach, as we believe that using both qualitative and quantitative data would enable an exploration of how the two methodological approaches could complement and nuance each other in understanding adolescents. Quantitative data come from the Ungdata study, a national youth survey covering almost all municipalities in Norway (Ungdata.no). Qualitative data were collected by the author in focus groups conducted in secondary schools in Innlandet County. Both datasets were analyzed separately and then brought together for analysis and comparison.

As Granic et al. [23] point out, in earlier research video games were defined broadly if at all. In this project, we draw on the understanding of video gaming of these authors, which comprises different types of electronic games played on PC/Mac, tablet, smartphone, or consoles like PlayStation, Xbox, and Nintendo Switch, which does not include video games used primarily for educational purposes. The most distinctive feature of video games compared to other media (e.g., television, movies, and books) is that they are interactive; gamers cannot passively surrender to a storyline of a game [23].

### 2.2. Quantitative Data

Ungdata is a standardized and quality assured survey covering various topics, such as school, health, family and friends, substance use, and leisure activities, including digital media use and video gaming. The Norwegian Social Research Institute (NOVA) administers Ungdata. The survey was first implemented in 2010 and is financed by the Norwegian Directorate of Health, Ministry of Children, Equality and Social Inclusion, and the Ministry of Justice and Public Security. Ungdata is available for grades 5–7 in elementary school (Ungdata junior) and for secondary school (Ungdata). The survey may be ordered free of charge every third year by municipalities. It comprises about 150 questions that are similar across all surveys. Schools may also choose an expanded version of the survey, with additional sets of questions on mental health, gaming behavior, and other topics. The students complete the self-reported digital questionnaire during one school lesson with a teacher present. Participation in Ungdata/Ungdata junior is voluntary and based on informed consent [24].

To show screen activity trends by age, we used the entire Ungdata junior and Ungdata sets used in 2010–2020. The analysis was restricted to participants with non-missing values in the variables of interest, which were gender, age, gaming on PC/console, gaming on smartphone/tablet and use of social media, with a total *N* = 504,553.

To analyze gender differences in secondary school adolescents regarding screen activity, we used data from 5607 8th to 10th graders (aged 13–16) in Innlandet County who participated in the Ungdata survey in the beginning of March 2020 before the national confinement due to the COVID-19 pandemic. The convenience sample includes 2779 (50.4%) boys and 2731 (49.6%), which represents about 45% of the secondary school students in Innlandet County. Categorical data were presented as counts and percentages. We used Chi-square tests to assess gender differences. *p*-values <0.05 were considered statistically significant. All analyses were conducted in SPSS Statistics for Windows, Version 26.0. (IBM, Armonk, NY, USA).

### 2.3. Qualitative Data

The qualitative data were collected using focus group interviews in September and October 2019. We focused on secondary school students aged 13–16 years, as this is the period when many girls lose interest in gaming while boys continue with it [14]. We recruited 25 students from two schools in Innlandet County. The school administration assisted us in recruiting participants and informing parents and students about the study. Participation was voluntary, and every interested student whose parents/guardians consented was eligible. Participants were sixteen boys and nine girls, roughly equally divided between grades 8 and 10. All participants had at least one friend or classmate that participated in the same group interview. We conducted five group interviews: one group of boys, one group of girls and three mixed. The composition of the groups was randomly, as students who wanted to participate were sent in groups of five by their teacher to the interview location, when it best suited the teaching situation. All participants spoke Norwegian as their mother tongue. The sample comprised adolescents with little video game experience, those who played on a recreational level, and those who had gaming as a passionate hobby.

An interview guide, including gaming-related questions from the Ungdata survey, directed the participants to talk about how they used video games, their experiences of friendship, intimacy, belonging, exclusion, or hate in gaming spaces, how gaming may support or constrain young people’s lives, their parents’ views on their video gaming, and the impact of gaming on their schoolwork, social relations, and mental health. The interview guide was pretested in a 9th grade class of 30 in a school where we did not conduct focus groups. The interview guide was modified after pretesting. Questions were not worded in a manner that directly targeted gender differences. The interview guide ensured consistency and flexibility in the approach to elicit the participants’ stories. Probes were open and specific to their comments.

Both researchers were present during all focus group interviews. The second author led the interviews, while the first author took notes and asked some follow-up questions. The interviews lasted 30–45 min and took place in a private room in the school with only participants and interviewers present. Confidentiality was ensured at the outset by informing participants that all identifying information would be removed from the data. All interviews were audiotaped and transcribed verbatim by the first author. The interviewers had the impression that all students spoke freely, but some were less talkative and gave shorter responses.

We conducted semistructured group interviews to generate interactive data and access everyday ways of talking about our research topics [25,26]. Focus groups are also a way to encourage a non-hierarchical context [27]. In the analysis of the interviews, we followed the guidelines for thematic analysis of Braun and Clarke [27]: 1. familiarizing yourself with your data, 2. generating initial codes, 3. searching for themes, 4. reviewing themes, 5. defining and naming themes, and 6. producing the report. We modified the six steps for our study as follows: Both authors listened to the recordings several times and read the transcripts closely for surface and underlying meaning, created codes to represent components of meaning and built themes by identifying patterns of meaning within and across transcripts. We used mind mapping to develop themes and relationships between concepts. We created primary codes individually, reviewed this coding together, and resolved discrepancies. We identified the following four themes: the significance of social media and gaming, views on playing and gaming, content and characters in video games, and acceptance and status in the peer group (Figure 1). To reflect the nuances of the identified themes, we enriched the results section with quotes from the participants, all of which have been translated from Norwegian.

### 2.4. Ethics

Quantitative data were obtained from an established database collected anonymously in the Ungdata surveys, which was approved by the Norwegian Centre for Research Data (NSD). Consequently, separate ethics approval for the present study was unnecessary. Informed consent was obtained from all students and their parents prior to the online survey. Separate informed consent was collected from the students participating in the focus groups and their parents. Students were also given verbal information about the study aim and their rights as participants at the beginning of the group interviews. Ethical approval for the qualitative part of the study was granted by NSD (NSD no. 882453).

## 3. Results

### 3.1. Survey Findings: Screen Activity

The quantitative analysis shows that more boys than girls in all age groups spend an average of one hour or more per day playing video games on a computer or console. The gender difference varies between 40% and 54%, depending on age. At age 13, video gaming on computers and consoles decreased by about 20% in both genders (Figure 2). Gaming on smartphones and tablets shows small gender differences. Here, the greatest difference was at age 11, with 45% of boys and 32% of girls gaming for one hour or more on average. At age 12, gaming on smartphones and tablets began to decline in both genders, and the gender differences were eliminated by age 14.

The analysis shows a gendered pattern in the adolescents’ use of social media (Figure 3). Between the ages of 11 and 19, more girls than boys were one hour or more per day active on social media. At age 11, the gender gap was about 10%, increasing to 30% at age 14, but declining to 20% at age 19.

Ungdata participants were also asked about video games, social relations, and friendship (Table 1). In Ungdata, 69% of boys, but only 16% of girls, found it very important to have contact with friends via gaming. More boys than girls (43% versus 12%) stated that they would have felt excluded if they did not play the same games as their friends. Most boys (79%) played online games with others at least one evening a week, while only 23% of girls did so. However, almost twice as many boys as girls (81% versus 43%) reported that the people they play games with are the same as those they meet in “real life”. All gender differences were statistically significant (*p* < 0.001 for all categories).

### 3.2. Interview Findings

#### 3.2.1. The Significance of Social Media and Gaming

The first theme reflected adolescents’ perceptions of how boys and girls were oriented towards different screen activities. In line with the survey findings, the participants associated video gaming with boys and social media with girls. One 9th grade boy said that boys and girls spend about the same amount of time on screen activities, but that boys spend most time on gaming and girls on social media. A girl in the 10th grade described how she and some friends used to text each other so much on social media that their parents had to make a rule forbidding the use of social media after 8.30 pm. She explained how video games and social media play different roles in her life: “I do think it’s fun to play video games, but now that I have Snapchat and stuff, I’ve kind of started talking a lot more with people there”. Both boys and girls mentioned using various social media platforms such as Facebook, Snapchat, and Instagram. Boys, however, did not view social media as socially significant as gaming. One boy in 8th grade stated: “I play a lot. That’s what I do most. I have social media too, but I’m not active there”.

Many of the boys used the term “gamer” about themselves. In their perspective, a “gamer” is a person who has video gaming as a passionate hobby. One 9th grade boy, who referred to himself as a gamer, described how gaming was an integral part of his life: “I play every day. Just now, it’s a lot of Minecraft, Grand Theft Auto, CounterStrike. I play with people from my class and sometimes with people I have gotten to know online”. The boys described video gaming as a broad medium, ranging from mobile app games (e.g., Candycrush) to large open world games (e.g., World of Warcraft). However, some genres and titles were described as more authentic video games than others. Boys described playing on a console or computer as “real” and “hardcore” gaming and playing on a tablet or a smartphone as “casual” gaming. When one boy mentioned that some girls at school play a horse game (“Howrse”) on tablets, one of the biggest gaming enthusiasts protested loudly: “Hell no, that’s not a real video game!”.

All of the girls we interviewed had some experience with video games. Most had one or more games on their smartphone or tablet, such as Candycrush, Roblox, Color Road, and Sims. Despite playing these games occasionally, the girls mainly expressed their relationship to gaming with comments like: “It’s just a pastime for me”; “It’s just something I occasionally do in vacations or when I’m traveling”; “I don’t sit down and do it for its own sake”; “You won’t find me sitting at a computer or Playstation”. Only one girl explicitly mentioned video games as her hobby. However, she expressed ambivalence towards video games and parts of gaming culture.

#### 3.2.2. Views on Playing and Gaming

The second theme related to how adolescents associated video gaming with different stages of life. Boys considered video gaming to be a central part of both childhood and youth culture. Many girls associated gaming primarily with childhood and younger children’s lifeworld. One 10th grade girl stated: “Many girls are insecure about themselves and can get affirmation that they’re good enough on social media. Some also think that gaming’s childish. Some games aren’t childish, of course, but on social media, you can talk more with friends”. Several girls reported having played more video games when they were younger, with friends or family members, but that their interest in gaming had gradually faded. One girl felt that video gaming was mostly about playing and having fun, but that social media was better suited to make and cultivate friendships. Some of the boys had noticed that the girls at school were more oriented towards social media and youth culture now than a few years ago. One boy in 9th grade explained: “Many of those typical ‘girl games’ aren’t so popular among the girls any longer. At age 9, or at least that’s what I think, all the girls want to be princesses, cute and stuff. But suddenly that stops, and they all want to be models on social media”.

From the boys’ perspective, video gaming belonged to both childhood and youth culture. Unlike the girls, some boys reported spending more time gaming now than when they were younger. Most of them had always enjoyed gaming and they found it hard to visualize a life without video games. One boy in 10th grade said: “A normal day for me is like this: I come home from school, I go to the computer, and I play video games (...) I play lots with friends. I’ve played soccer before, but I’ve become a gamer. I’d probably survive without gaming, but it’s so cool!”. Video gaming was emphasized as fun and entertaining by several of the boys. One boy in 10th grade used to phrase “digital playing” to describe his relationship to gaming and mentioned the popular video game Minecraft, which he described as “Lego on the screen”.

Some boys also explained that gaming was more than pure play and fun: they connected gaming to learning, education, and careers. Several boys envisioned a future where video games would be part of their lives, not solely as a passionate pastime but also as a possible career, such as a professional video gamer (eSports athlete), streamer, or game developer. The boys mentioned several schools that offered gaming programs. Some boys were deeply interested in eSports, and they thought it was cool that Fortnite had a Norwegian world champion.

#### 3.2.3. Content and Characters in Video Games

The third theme related to video game content and characters. Both boys and girls were critical of gender representation in video games. One 9th grade girl expressed it this way: “The characters in video games are mostly guys. There are lots of war games. I don’t play any video games where the main character is a man”. Although she described gaming as her hobby, she was ambivalent about video games because she felt that there were so few “good female characters”. Another girl commented that female characters are often portrayed as helpless and in need of being rescued, mentioning the princess character in the Super Mario Bros games.

Both girls and boys felt that video games were addressed more to males than females. One 10th grade boy explained: “It’s nice to have a choice. Then maybe girls will feel more welcome in video gaming (...) I play a female character in Destiny 2. All the male characters in the game look completely hopeless. They all look like they’re in a midlife crisis.” The interviewees wanted more nuanced and varied game characters. They thought more gender balance would improve the games and that the introduction of more female characters could make girls feel more attached to gaming.

#### 3.3.4. Acceptance and Status in the Peer Group

The fourth theme concerned the greater social acceptance and higher status of gaming among boys than among girls. The boys described how gaming was an essential setting for experiences of togetherness in free time and school time. One 10th grade boy said: “I play with classmates and sometimes with people I’ve got to know online (…) We have an iPad at school, and it’s tempting to play a game in class. You get a black mark if the teacher catches you, but my friends and I take that risk”. Several boys said that they got to know others better and made friends in online games. Some mentioned that they kept in touch with family members and former neighbors or classmates by playing online together. “In games like Minecraft or Super Mario Maker, you can build something together while talking about other things”, one of the boys explained. Another boy said that he and some classmates used to play Minecraft after school. He added: “I’d have felt left out if I couldn’t play with my friends”.

Besides enjoying playing video games themselves, several boys reported often spending time on YouTube and Twitch watching others play video games. They found this just as natural and meaningful as watching soccer or other sports on TV. The most popular games among boys included World of Warcraft Classic, Fortnite, FIFA, Call of Duty, and Counter Strike. When we asked if they sometimes played with or met girls in these games, one boy answered: “I’d be amazed if I met a girl in CounterStrike. There are really not that many girls in those games”. For the girls, social media had greater social significance than video games in the peer community. One girl described how she began to think twice about gaming when she was 12: “When people say gamer, I think mostly of boys really (...) I used to play a boxing game. That was before I got social media. Every time I got a visit, I tried to get them to play, but (...)”. This lack of interest in video gaming from her friends colored her view of video games. Girls found it difficult to find other girls to play with or talk to about games.

## 4. Discussion

Our study found pronounced gender differences in the screen activity of adolescence. The statistical analysis showed that boys spent significantly more time than girls in all age groups playing video games on consoles or computers. Regarding online gaming, most boys reported playing at least once in the past week, compared to 23% of girls. This finding concurs with other studies from Norway [14], Germany [28], and the USA [29], which report that a greater proportion of boys play video games. However, as our analysis indicated, it is useful to distinguish between video games played on consoles and computers and those played on smartphones and tablets, as the latter did not show gender differences. While many boys preferred large role-playing games (MMORPGs) and action-adventure games, girls seemed to prefer playing games casually for shorter periods. One way to understand these gendered patterns in gaming behavior could be that video games serve different purposes in the lives of 11–19-year-old boys and girls. This interpretation is supported by previous research suggesting gender-specific motivations for playing video games: while many girls and women prefer games suitable for relationship maintenance, boys and men are often more interested in complex and competitive gameplay [30,31]. Gender is a fundamental human characteristic that affects the societal roles and behavior of boys/men and girls/women in different private and public spheres [32,33]. The gender intensification hypothesis proposed by Hill and Lynch [34] assumes that adolescent boys and girls are confronted with increased pressure to conform to culturally sanctioned gender roles at the beginning of this life phase. So-called appropriate gender roles are conveyed by parents, peers, educators, and the media. In light of these factors, adolescents are thought to become more differentiated in their gender-role identities, which might influence their adult roles as women and men. This gender intensification hypothesis may explain how gender differences (here in screen activity) emerge or intensify during adolescence. Despite evidence that the gaming landscape has included more female gamers in recent years, our informants mainly perceived video gaming as an activity for adolescent boys. Our interviewees’ statements suggest that gaming demonstrates a particular hierarchy. At first glance, “gamer” could appear to be an open and inclusive identity term for anyone involved in gaming activities. However, our informants, especially the boys, did not recognize games on smartphones or tablets as “real” video games. To be perceived as a real gamer, one has to play on a console or computer. Playing games on a tablet or smartphone was often described as something people do “on the go”, while waiting for the bus or during meals. This might be because gamer stereotypes are still gendered. Morgenroth and colleagues show in a recently published study that both negative and positive facets of masculinity such as poor social skills, competence, aggressiveness, and agency are associated with male gamer stereotypes [35].

Further, the analysis showed that the adolescent boys and girls had different views on playing and gaming. Previous studies have reported that teenage girls are often keen [36] to create their identities around new practices associated with adolescence, while distancing themselves from childhood-associated play activities [37,38,39,40]. Our findings are similar in that many girls linked video games to childhood. The girls frequently described video games as less socially diverse than social media, which was often felt to be more active and engaging. Our results also support the gender intensification hypothesis. As hypothesized by Hill and Lynch [34], gender disparities increase during adolescence, which our study confirms: at the age of 13–14 years, girls lose interest in gaming and focus more on social media, while boys identify themselves as gamers, as shown by both the qualitative and quantitative data. Both genders considered that being active in social media and gaming respectively was vital to socialize with their peers.

Another factor that the adolescents eagerly discussed as a gender barrier in gaming was the predominantly male/masculine orientation of many video games. It is well documented that there are far more male than female game characters [41] and that female avatars are often portrayed in sexualized supporting roles with sexy clothing and large breasts [42,43]. A literature review on gender differences in online gaming [13] found that the overall masculine content and characters in video games could be a significant barrier to girls/women embracing video gaming, which resonates with our findings. Although there might be an increased awareness of gender and sexuality representations in video games today, among both players and developers, our study shows that some girls feel overlooked or excluded by the video game community. Further, several participants noted that various negative experiences could create gender barriers in video gaming. Both genders seemed to agree that gaming culture is more hostile and toxic towards females than males. This pattern conforms with a narrative literature review on the role of women in gaming culture, which concluded that female gamers appear to require coping strategies to handle online harassment, as video games are associated with stereotypical male characteristics, such as aggression and an interest in sexualized content [44]. As suggested by Fox and Tang [45], the persistent perception of gaming (especially on consoles and computers) as a male territory potentially hostile to females could explain the gender gap in gaming behavior. Some studies suggest that online games’ anonymity and competitiveness increase the likelihood of toxic behavior [46]. In a Norwegian study, Ask and Svendsen [42] found that half of the female players in their sample had actively concealed their gender to avoid sexual harassment in online games. They might choose male avatars and gender-neutral names or avoid talking to other players to hide their gender identity. Cote [47] pointed out that such strategies have significant downsides since ignoring toxicity or brushing it off allows it to persist. When hiding gender identity becomes an integral part of gaming, both the gaming experience itself and the sense of belonging to the gaming community are weakened [17]. However, several video game companies and developers are planning to create more healthy online communities. For instance, Sony has announced a new feature for PlayStation 5 that allows players to use voice chat to quickly report verbal harassment to Sony customer support [48].

For many young people today, especially boys, gaming occupies most of their waking hours out of school. For many of the boys, video games were a crucial aspect of their social life. They emphasized that gaming was deeply integrated into their friendship and peer groups. Playing video games with friends allowed boys to communicate their thoughts, feelings, and worries in a play-like context. For some boys, video games were also a way to keep in touch with former classmates or distant relatives, as also described in other research [49].

The quantitative analysis supports the interviews in finding that gaming is an essential social tool in male adolescents’ lives. Gaming is a way to connect with friends and find like-minded peers, especially for boys. It is important not to underestimate or condemn virtual friendship as less real and genuine. Although the use of digital tools that mediate our social interactions, such as online gaming and social media, help to maintain communication during breaks in physical presence [50], researchers disagree on whether such means of communication reduce the quality of relationships and empathy for others [51,52]. In a previous article, we argued that video games may provide essential building blocks when young people, particularly boys, form their identities and peer cultures [53]. For the girls, social media had greater social significance than video games in the peer community. Girls could have difficulty in finding other girls interested in gaming and in joining a gaming community. They were less likely than boys to describe video games as a social tool.

### Strengths and Limitations

This study focused on gaming an understanding of gaming behavior in adolescents in a gender perspective. We applied a mixed methods approach to obtain a broad picture of adolescents’ video gaming practices and experiences. There are several important limitations to the study findings to be addressed. Firstly, one challenge that can emerge when applying a gender perspective is to overemphasize gender differences. Gender and other social identities, such as age, ethnicity, sexuality, religion, and class, form part of a complex interrelationship [32]. Our interview sample included both genders, different ages, and varying degrees of experience of playing video games. However, our sample has a disproportionate gender composition (16 boys and 9 girls). Further, it did not include any participants of immigrant origin. The inclusion of a more diverse sample might contribute to a deeper understanding of gaming patterns between and within the gender groups. Research has also shown that personality traits, such as the different components in the Big Five model, may influence gaming behavior. For example, Markey and Markey found that persons who score low on agreeableness, low on conscientiousness and high on neuroticism may react more negatively to violent video games than other persons [54]. Future research should explore how gender intersects with other social identities and individual characteristics in video gaming patterns. Secondly, this study aimed to achieve an overall understanding of gamers’ practices and experiences. Apart from distinguishing between video games played on consoles and computers and those played on smartphones and tablets, this study did not explore possible nuances between different game genres or titles. As pointed out by McLean and colleagues, video games can have unique environments and very different user bases [55]. Consequently, the findings from this study may not be generalizable to specific game genres or titles. In line with Bopp et al., future research could investigate whether certain games and gaming communities are more inclusive/exclusive than others [56].

Besides these limitations, our study also possesses some strengths. It focused on the gender perspective of gaming behavior in adolescents, which has been absent in much previous research [17]. While the survey provided comprehensive data about video gaming behavior among Norwegian adolescents, the group interviews elicited the participants’ perspectives, language, and concepts, thus providing the study with high ecological validity [57]. Another strength in this study is that the participants in the focus group were very heterogeneous when it comes to their personality, appearance, and degree of activity, as far as we could capture this during the interview situation, which provided us a diverse insight in their adolescence everyday life. The results from this study will complement the clinical and psychological approaches that dominate the research field around video games [58], and provide empirical knowledge for public debate about gender, equality, and social inclusion, which is an urgent public health problem. Exploring video gaming from the gamers’ perspective and identifying protective and harm-reducing factors are important steps towards developing interventions to promote more healthy gaming behavior and communities.

## 5. Conclusions

The present study found significant gender differences in the video gaming of Norwegian 11–19-year-olds. The quantitative part showed a high proportion of gamers among boys regardless of age, but less gaming among girls with increased age. The qualitative part identified four themes that may explain how gender effects adolescents’ practices and gaming experiences. Gender differences in gaming behavior and experiences are not necessarily problematic. However, our study found that adolescent girls have a more limited scope in gaming. The finding of gendered barriers to participation in gaming activities must be explored further in a larger population.

## Figures and Tables

**Figure 1 ijerph-18-06085-f001:**
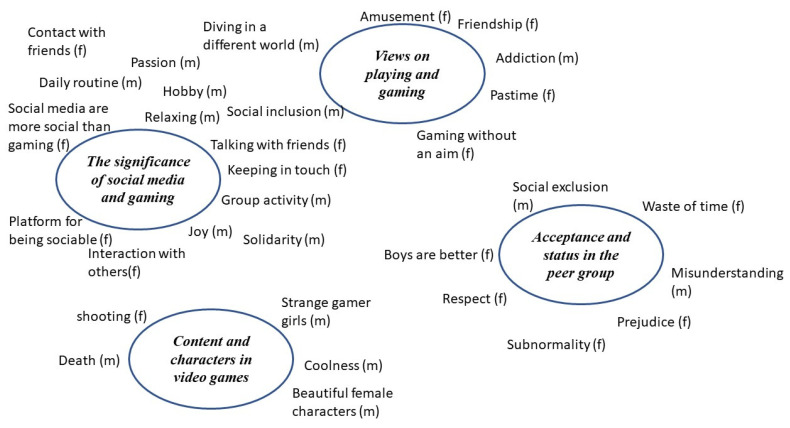
Themes generated by mind mapping. (m) = mentioned by male participants; (f) = mentioned by female participants.

**Figure 2 ijerph-18-06085-f002:**
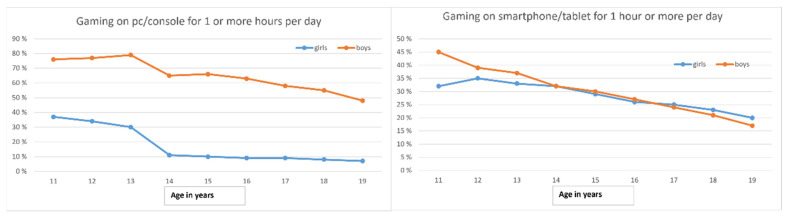
On the left: Percentage of students gaming on PC/console one hour or more per day, by age and gender. On the right: Percentage of students gaming on smartphone/tablet one hour or more per day, by age and gender. Gender differences are significant with a *p*-value of <0.001 in all age groups.

**Figure 3 ijerph-18-06085-f003:**
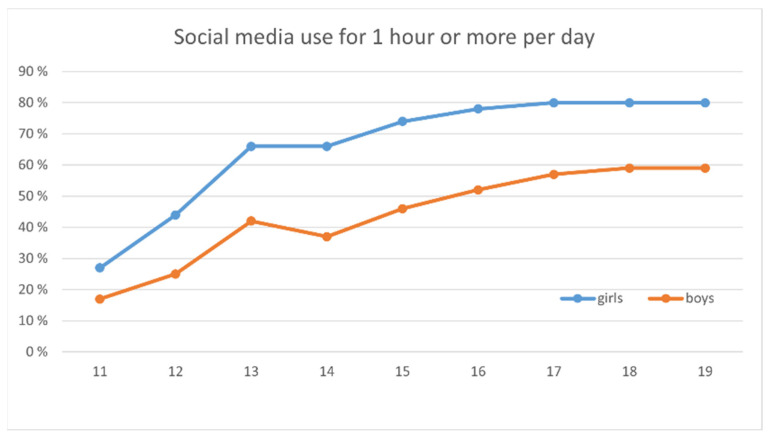
Social media use (one hour or more per day) by age and gender. Gender differences are significant with a *p*-value of <0.001 for all age groups.

**Table 1 ijerph-18-06085-t001:** Gaming and friendship by gender. Numbers and percentages of those who answered yes.

Variable	Boys		Girls		
	%	*N*	%	*N*	*N* Total
It is very important for me to have contact ^**a**^ with friends through gaming	69	1715	16	373	4850
I would feel excluded if I did not play the ^**b**^ same video games as my friends	43	1054	12	289	4805
The people I play video games with are the ^**c**^ same people I meet in “real life”.	81	1972	43	978	4712
I play online video games with others at ^**d**^ least one evening a week	79	2078	23	616	5268

**^a^** χ^2^(1) = 1428.9, *p* < 0.001, *N* = 4850; **^b^** χ^2^(1) = 568.8, *p* < 0.001, *N* = 4805; **^c^** χ^2^(1) = 723.1, *p* < 0.001, *N* = 4712; **^d^** χ^2^ (1) = 1621.5, *p* < 0.001, *N* = 5268.

## Data Availability

Data available on request due to restrictions. The data presented in this study are available on request from the corresponding author. The data are not publicly available due to protection of data privacy.

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
