# Peer review of "Are There Differences in Video Gaming and Use of Social Media among Boys and Girls?—A Mixed Methods Approach"

_ijerph, 2021, doi:10.3390/ijerph18116085_

Round 1
Reviewer 1 Report
Detailed comments are in the attached document
Author Response
Dear reviewer 1,
Thank you very much for providing us with the opportunity to revise our manuscript. Many thanks for your constructive suggestions on how to improve the manuscript. Below you will find our point-by-point responses to your questions and concerns. All major changes are highlighted in the revised manuscript in blue.
Comments of the reviewer |
Our reply |
1. Line 26 and 41: highlight of apostrophe
2. Line 46-52: The years from the papers published may not support these statements. Line 46: Paper from 2017, Line 50: Papers from 2010,2014,2018,2020 3. Line 53, 77, 92. 96: avoid " 's "
4. Line 149/150: highlighted text
5. Figure 1: explicitly explain letters in parenthesis. This image is very interesting, but the distribution of the information can be improved
6. Line 187: is this statement empirical?, or there is an statistical analysis behind?, if so, include it
7. Page 6 and 7: highlight of apostrophe
8. Line 319: highlight of “s”
9. Line 451: highlight of “four themes”
10. References: highlight of year |
1. Thank you very much for your review. Before submitting our article to the IJERPH we have sent the article to professional language editing. After consulting the editor again, we have changed the apostrophe “s” to a “.. of..” formulation 2. Good point, we agree. The references from 2010 and 2014 have been substituted with newer research (2018 &2020), so the statements in line 46-52 are now better supported. 3. We have changed the apostrophe “s” to a “.. of..” formulation 4. We have changed the wording of the phrase 5. Thank you for pointing this out. Letters in parenthesis have been now explained
6. This statement is a result of an eyeball inspection of figure 2, as described in the text. This is standard practice in similar analysis of the longitudinal study “Ungdata” and “Ungdata junior”. The figures are supporting material and therefore detailed statistical analysis are not presented. Nevertheless, we have added the p-value under the figure. 7. We would like to keep the apostrophe here since we cite the adolescents. We believe that using “it’s” instead of “it is” reflects better how the students talk, although the citations have been translated from Norwegian to English. 8. we have changed the apostrophe “s” to a “... of…” formulation 9. Sorry, but we do not understand this highlighting 10. Thank you for highlighting this. We have added a space between the year and the rest |
We hope that you will find the revised version suitable for publication in the International Journal of Environmental Research and Public Health.
Yours sincerely Marja Leonhardt
Reviewer 2 Report
Overall comments: This is a really well written manuscript. The study reported on is well designed and the observations very interesting. The authors do a good job obtaining information from an age range of adolescents that are often overlooked in research activities. Kudos to the authors for a good and impactful article for the peer reviewed literature. I do have minor comments which will make the manuscript even better.
Specific comments:
- Line 10, verb should be “are” instead of “is”.
- Line 15, grammatical correction needed. Text should “…games up to 5 times more than girls..”.
- Line 15, do girls spend more time on social media? Please clarify.
- Line 20, it might be useful to list what the different “gender-related experiences” are that result in how girls and boys feel about gaming. Was that information captured in the focus groups?
- The authors may want to consider including text that explores the activities/behaviors/interactions that cause boys and girls to feel differently about gaming. What are the social and societal implications of boys and girls attitudes towards gaming.
- Lines 26-27, reference needed for that sentence.
- Lines 34-36, reference needed for that sentence.
- Lines 59-64, is there any scientific evidence for why gaming interest is increasing in girls? It might be useful to include that information here in this paragraph. Also, a 13% change in the number of female gamers is quite large and should be more prominent in the discussion.
- Line 88, did the authors conduct the focus groups? Or were the data from the focus groups available to this research group for analysis and reporting? Please clarify.
- Line 115, the number of participants is listed as n=5607. What is the percentage of males and females in that number? How does this compare to the total number of children in this age range in the county? Should it be clarified that this is a convenience sample or is it nationally representative of the children aged 13-16 years in all of Norway? Please consider adding more detail describing the number of participants.
- Lines 181-182, consider including information on the numbers of children in each age category as a function of sex. Then Figure 2 makes more sense.
- Figure 2, please label the x-axis. Please explain what the N values are in the title of each panel in Figure 2. Is this the number of children who report gaming on each type of device? And, why doesn’t it add to the total participant N?
- Figure 3, are these differences statistically significant? What does the N value represent in the title of the figure? Please explain. Please label the x-axis.
- Line 205, statistically significant at what p value?
- Line 210, verb should be past tense.
- Line 244, verb should be past tense.
- Line 266, please edit to read “… in 10th grade used the phrase…”
- Line 277, verb should be past tense.
- Line 293, verb should be past tense.
- Line 319, please delete the word “authors” at the beginning of the sentence.
- Discussion and Strengths and Limitations sections, is there anything about the children and adolescents that responded to the survey questions that would influence the results reported? What about activities and behaviors? Introverted versus extroverted personality types? This appears to be a convenience sample – might that influence the results observed? Consider how that should be addressed as both a strength and limitation.
Author Response
Point-to-point response letter- reviewer 2
TITEL: Are there differences in video gaming and use of social media among boys and girls? – a mixed methods approach.
Oslo, 2nd of June2021
Dear reviewer 2,
Thank you very much for providing us with the opportunity to revise our manuscript.
Many thanks for your constructive suggestions on how to improve the manuscript. Below you will find our point-by-point responses to your questions and concerns. All major changes are highlighted in the revised manuscript in blue.
Reviewer 2
Comments of the reviewer |
Our reply |
1. Line 10, verb should be “are” instead of “is”. 2. Line 15, grammatical correction needed. Text should “…games up to 5 times more than girls..”. 3. Line 15, do girls spend more time on social media? Please clarify. 4. Line 20, it might be useful to list what the different “gender-related experiences” are that result in how girls and boys feel about gaming. Was that information captured in the focus groups? 5. The authors may want to consider including text that explores the activities/behaviors/interactions that cause boys and girls to feel differently about gaming. What are the social and societal implications of boys and girls attitudes towards gaming. 6. Lines 26-27, reference needed for that sentence. Lines 34-36, reference needed for that sentence. 7. Lines 59-64, is there any scientific evidence for why gaming interest is increasing in girls? It might be useful to include that information here in this paragraph. Also, a 13% change in the number of female gamers is quite large and should be more prominent in the discussion.
8. Line 88, did the authors conduct the focus groups? Or were the data from the focus groups available to this research group for analysis and reporting? Please clarify. 9. Line 115, the number of participants is listed as n=5607. What is the percentage of males and females in that number? How does this compare to the total number of children in this age range in the county? Should it be clarified that this is a convenience sample or is it nationally representative of the children aged 13-16 years in all of Norway? Please consider adding more detail describing the number of participants. 10. Lines 181-182, consider including information on the numbers of children in each age category as a function of sex. Then Figure 2 makes more sense.
11. Figure 2, please label the x-axis. Please explain what the N values are in the title of each panel in Figure 2. Is this the number of children who report gaming on each type of device? And, why doesn’t it add to the total participant N? Figure 3, are these differences statistically significant? What does the N value represent in the title of the figure? Please explain. Please label the x-axis. 12. Line 205, statistically significant at what p value? 13. Line 210, verb should be past tense. Line 244, verb should be past tense. Line 266, please edit to read “… in 10th grade used the phrase…” Line 277, verb should be past tense. Line 293, verb should be past tense. Line 319, please delete the word “authors” at the beginning of the sentence. 14. Discussion and Strengths and Limitations sections, is there anything about the children and adolescents that responded to the survey questions that would influence the results reported? What about activities and behaviors? Introverted versus extroverted personality types? This appears to be a convenience sample – might that influence the results observed? Consider how that should be addressed as both a strength and limitation.
|
1. Thank you very much. We have changed it. 2. Thank you for pointing this out. We have corrected the sentence.
3. and 4. Thank you for this suggestion. Our abstract comprises already the maximum number of letters which are allowed in the abstract. This, we decided avoiding going into detail here.
5. Thank you for this comment. The focus groups gave us some insight in how girls and boys feel differently about gaming. We present the result concerning this topic in point 3.2.2 Views on playing and gaming and 3.2.3. Content and characters in video games and the discussion 6. We have added some references.
7. Thank you for pointing this out. We agree that this increase is both substantial and interesting. However, the research on female gaming is still scarce and we haven´t found much research investigating why the numbers of female gamers are on the raise. That is why we also in the discussion only can highlight some possible factors behind this trend (e.g. that the gaming industry is turning more to girls/women with more inclusive game content and characters and that gaming may have become more socially accepted). 8. Thank you for this comment, we clarified that the authors conducted the focus groups.
9. We analyze in our study sample of secondary school students in the Norwegian county Innlandet. Unfortunately, statistics Norway only publishes numbers in the average 13-15 and 16-17 which is not the same as in our sample. Therefore, we cannot provide national numbers. But we have added a more detailed description of the sample in line 112-120 and the following.
10. Thank you for this suggestion which we considered carefully. We believe that the figures contain enough information and facilitate the readability of the manuscript. Although we did add clarifying information under the figures. 1. Thank you for pointing this out. We have labeled the figures as suggested. We understand, that reporting the total N is confusing. Therefor we have restricted the analysis to participants with non-missing values in the variables of interest (see line 113).
2. We have added the p-value in parenthesis 3. Thank you for pointing this out. We have corrected the grammar and highlighted changes in blue
14. Thank you very much for this comment. We now discuss the effect of the sample and personality traits more elaborately under “strength and limitations”.
|
We hope that you will find the revised version suitable for publication in the International Journal of Environmental Research and Public Health.
Yours sincerely
Marja Leonhardt
Reviewer 3 Report
Nowadays video games are an extremely widespread leisure activity, and their popularity only grows every year. It can be assumed that over the past year, in a pandemic situation, this trend has only intensified. More and more people are getting involved in video gaming, which is becoming not only the main form of recreation but also the leading way of socializing, especially among adolescents and children.
In this regard, the study presented by the authors is relevant and important, allowing to obtain new data on the role of video games in the life of adolescents. The authors focus on gender differences in video gaming behavior. In addition, the undoubted advantage of the study is the combination of quantitative and qualitative approaches, as well as the use of a large database of a standardized and quality assured national survey.
The study found significant gender differences in the video gaming of adolescents and showed a relationship between gender and adolescents’ practices and gaming experiences.
Although the manuscript is recommended for publication, I would like to make a few comments:
1.There are different types of video games. It seems that the revealed gender differences primarily relate to strategy and action games. However, does this apply to a large number of other types of games, such as educational ones? Perhaps video games need to be differentiated to better define gender differences in their consumption and to identify their role in the life of adolescents.
2. What is the reason for the disproportionate gender composition of focus groups (sixteen boys and nine girls)? Could this have influenced the results obtained?
3. The detailed presentation of the implementation procedure of the qualitative approach in the paper should be noted, however, clarification is required. The authors point out that focus groups were conducted both mixed and separately for boys and girls. What was the purpose of this differentiation? And how was this reflected in the results? In general, it seems that in order to achieve the aims of the study, separate mind mapping for males and females should be constructed. This would perhaps reveal different themes in boys and girls and get a more complete picture of gender differences in gaming behavior.
4. It seems that the word 'influence' needs to be used more cautiously in describing the study and its results, as the statement of influence is usually the result of an experiment.
5. Chi-square test data are not provided.
Author Response
Dear reviewer 3, please see the attachment.
